# Impact of the National Vaccination Strategy on the Prevalence of *Streptococcus pneumoniae* and Its Serotypes Among Clinically Healthy Children Under Six Years of Age During and After the COVID-19 Pandemic

**DOI:** 10.3390/vaccines13060634

**Published:** 2025-06-12

**Authors:** Ivelina Trifonova, Victoria Levterova, Ivan Simeonovski, Magi Ivanova, Nadia Brankova, Todor Kantardzhiev

**Affiliations:** National Centre of Infectious and Parasitic Diseases, 26 Yanko Sakazov Blvd., 1504 Sofia, Bulgaria; ivanos@abv.bg (I.S.); magiivanova-96@abv.bg (M.I.); nbrankova@abv.bg (N.B.); todorkantardjiev@gmail.com (T.K.)

**Keywords:** *Streptococcus pneumoniae*, 19A, 6C, PCV10, serotypes, vaccination, healthy children, antibiotic-resistant serotypes

## Abstract

Introduction: An effective vaccination strategy requires monitoring serotype changes by geography and age. This study analyzed *Streptococcus pneumoniae* serotypes in healthy children under 6 years of age vaccinated with PCV10 in Bulgaria from October 2021 to May 2025. Methods: A total of 569 children were screened for the *lytA* and *cpsA* genes viareal-time polymerase chain reaction (real-time PCR). Positive samples were typed using relevant kits, and 76 serotypes/serogroups of *S. pneumoniae* were identified. Results: Nasopharyngeal swabs from 232 children (40.8%) were found to carry *S. pneumoniae*, and a total of 255 serotypes were detected, with 19B/19C (17.2%), 6C (10.7%), and 15B/15C (9.8%) being the most prevalent. Of these, 91 serotypes (15.9%) were included in at least one vaccine, while the remaining 164 serotypes (25.4%) were not. The carriage rate reduced to 22% in 2023 but increased to 47% in 2024. Overall, younger children had lower carriage rates (*p* < 0.05), with serotype 6C being more common in children under 12 months of age (25%). Approximately 9.1% of pneumococcal carriage cases involved co-detected serotypes, with significantly higher co-detection rates for 19B/19C, 15B/15C, 10B, 10F/C, 23B, 7C/40, 23A, and 24A compared with mono-detection rates (*p* < 0.05). Conclusions: 19B/19C, 6C, 15B/15C, and 19A were identified as the main serotypes. Children over 3 years of age were also more likely to carry multiple pneumococci. These findings emphasize the need to reassess childhood vaccination strategies to curb the spread of antibiotic-resistant serotypes.

## 1. Introduction

*Streptococcus pneumoniae* represents a significant public health challenge worldwide. It is one of the most common bacterial pathogens responsible for invasive disease in both children and adults [1]. According to the World Health Organization (WHO), an estimated 1.6 million people died from pneumococcal disease in 2011, with between 0.7 and 1 million of these deaths occurring in children under 5 years of age, predominantly in developing countries [2]. The high mortality rates can be explained by the wide global distribution of *S. pneumoniae* and its ability to colonize the nasopharynx without causing symptoms [3,4]. This bacterium can spread rapidly, especially in close-knit communities, through airborne transmission via droplets or direct contact with carriers [5].

*S. pneumoniae* has a complex relationship with its human host. These bacteria are highly adapted companions that primarily inhabit the mucosal surfaces of the upper respiratory tract of carriers, facilitating their transmission [1,6]. However, they can also cause severe disease when certain bacterial and host factors allow them to invade normally sterile areas such as the middle ear, lungs, bloodstream, and meninges [7]. *S. pneumoniae* can both evade and exploit host inflammatory and immune responses, which determine the processes of transmission, colonization, and invasion [1,8]. These features in pneumococcal pathogenesis make it an important target of epidemiological research aimed at developing strategies to reduce transmission and prevent complications associated with pneumococcal infections.

The polysaccharide capsule of this bacterium is its most important pathogenic factor [3]. It is a major virulence and antigenic component in *S. pneumoniae* and plays a crucial role in the bacterium’s ability to colonize hosts. Pneumococcal serotypes are classified based on variations in the structure of their polysaccharide capsules [9,10]. Since the introduction of pneumococcal conjugate vaccines, there has been a noticeable decrease in the asymptomatic carriage of vaccine-targeted serotypes of this pathogen [11]. Over 100 different pneumococcal serotypes have been identified, varying in frequency, disease manifestation, and antibiotic resistance [12,13]. Many factors, including the introduction of new pneumococcal conjugate vaccines (PCVs) and the use of antibiotics, may influence the spread of different *S. pneumoniae* serotypes [14]. These serotypes vary in frequency, clinical manifestations, and antibiotic resistance. The introduction of new conjugate vaccines and antibiotic usage can influence the replacement of these serotypes [15]. The first pneumococcal polysaccharide vaccine was tested in 1944 [16]. Interest in vaccines declined after the discovery of penicillin, but the need for prevention led to the introduction of the 23-valent pneumococcal polysaccharide vaccine (PPSV23), covering 23 serotypes, in 1983 [17]. PPSV23 primarily induces a B-cell-mediated immune response, making it ineffective for infants and toddlers at high risk of pneumococcal disease. To address this, pneumococcal conjugate vaccines (PCVs) were developed to elicit a T-cell-dependent immune response, inspired by the success of the *Haemophilus influenzae* type b vaccine [18]. The first pneumococcal conjugate vaccine (PCV), known as the 7-valent PCV (PCV7), was introduced in 2000 and demonstrated high effectiveness in children. This was followed by two additional higher-valency PCVs: the 10-valent PCV (PCV10), introduced in 2009, which included all the serotypes covered by PCV7 plus three additional serotypes (1, 5, and 7F); and the 13-valent PCV (PCV13), launched in 2010, which included all the serotypes in PCV10 plus three more (3, 6A, and 19A) [19]. These three vaccines—PCV7, PCV10, and PCV13—are part of national immunization programs (NIPs) for children in many countries worldwide.

In 2010, Bulgaria’s Ministry of Health approved and instituted mandatory vaccination with the 10-valent pneumococcal vaccine (PCV10) in children. Immunization against pneumococcal disease is administered in four doses, scheduled at 2, 3, 4, and 12 months after birth. From 2011 to 2023, vaccine coverage was reported to be around 90%, according to data from the National Center for Infectious and Parasitic Diseases [20]. Developing an effective vaccination strategy for each country requires monitoring changes in the serotypes that colonize the population and analyzing geographic and age-specific distribution patterns. Despite the critical need to select the right vaccination approach, the existing literature does not sufficiently address the epidemiology of emerging serotypes. To evaluate the long-term impact of vaccination, we aim to conduct an epidemiological analysis of the different *S. pneumoniae* serotypes carried by healthy children up to 6 years of age who have been vaccinated with PCV10.

## 2. Materials and Methods

### 2.1. Study Population

This study focused on vaccinated, healthy children aged 5 months to 6 years. It was conducted from October 2021 to May 2025, providing data on the carriage and distribution of *S. pneumoniae* serotypes in Bulgaria. Samples for this study were collected after obtaining informed consent from the parents.

Nasopharyngeal samples were collected using a flexible, sterile swab. The samples were collected by trained personnel during planned home visits. Sample collection occurred year-round, with secretions collected from children up to 6 years of age in kindergartens and by general practitioners across 15 of 28 districts in Bulgaria: Sofia, the Sofia region, Yambol, Pernik, Montana, Shumen, Burgas, Blagoevgrad, Gabrovo, Plovdiv, Kardzhali, Stara Zagora, Kyustendil, Pleven, and Vidin.

### 2.2. Participant Selection Criteria

−Healthy children aged 2 to 6 years who attended childcare facilities, having completed a full vaccination course of PCV10 (3 + 1 doses).−Healthy children aged 5 months to 6 years who did not attend childcare facilities but had received at least one of the mandatory doses of PCV10.

### 2.3. Sample Collection

Nasopharyngeal secretions were collected using two swabs: one with eSwab transport medium (Copan, Italy) for microbiological examination on nutrient medium and one dry sterile swab for DNA isolation. The samples were transported to the National Reference Laboratory (NRL) for Molecular Microbiology at the National Centre of Infectious and Parasitic Diseases in Sofia, Bulgaria. Samples from children under 11 months were collected before administration of the fourth vaccine dose. No additional biological samples were included in this study, and the participating children remained anonymous; only data on year and month of birth, gender, kindergarten attendance, and vaccination status were used.

### 2.4. Exclusion Criteria

The study excluded patients who exhibited symptoms of respiratory infections and those older than 6 years of age, as well as individuals who were unwilling or unable to provide consent. Additionally, any samples received in the laboratory that did not comply with the specified criteria for the transportation and storage of nasopharyngeal samples were also excluded.

### 2.5. Detection of S. Pneumoniae

#### 2.5.1. Classical Detection Method

The collected samples were cultured on Columbia CNA agar with 5% sheep blood, supplemented with an optochin disc to differentiate pneumococci from other flora, at 35 ± 2 °C for 18–24 h in an aerobic atmosphere enriched with carbon dioxide at an elevated concentration (8–10%). In the presence of *S. pneumoniae*, it was isolated, subcultured, and subsequently used for DNA extraction and strain storage [21]. 

#### 2.5.2. Molecular Methods

For direct DNA isolation from swabs containing nasopharyngeal secretions, a protocol using 5% Chellex [22] ion exchange resin was followed, in accordance with the standard procedures of the National Laboratory of Molecular Microbiology. Samples were screened for the *lytA* (in Appendix A) and *cpsA* genes using real-time PCR [23]. Samples positive for both genes were further typed using PCR and subsequent allelic hybridization for 76 serotypes and serogroups. Samples positive only for *lytA* were classified as non-capsular, and pneumococcus typing was not performed for these samples.

#### 2.5.3. Typing of *S. Pneumoniae* Isolates

Samples positive for both genes were typed using PCR and subsequent allelic hybridization using the commercial kit “*S. PneumoStrip*” from Operon, allowing easy and rapid single amplification identification of 76 serotypes/serogroups of *S. pneumoniae*. Among these serogroups are those included in the currently available vaccines.

The following serotypes/serogroups were included:

Band A: 1, 3, 4, 5, 6A, 6B, 6C, 6D, 7F/7A, 9A/9V, 14, 18A, 18B/18C, 18F, 19A, 19F, 23A, 23B, 23F.

Band B: 2, 8, 9N/9L, 10A, 10B, 10F/C, 11A/D, 11B, 11C, 11F, 12A/46, 12B/44, 12F, 15B/15C, 17F, 20, 22F/22A, 33F/33A, 37.

Band C: 7B, 7C/40, 15A, 15F, 16F, 19B/19C, 21, 25A/25F, 38, 24A, 24B/24F, 31, 32A/32F, 33B/33D, 33C, 35A, 35C, 35F, 47F, 41A, 41F.

*LytA* and *cpsA* were also identified as control genes during the hybrid analysis.

### 2.6. Statistical Analysis

We used a 2 × 2 contingency table to analyze the data, using the number of patients by two or more experimental factors, and dividing participants into appropriate categories. A Fisher’s exact test was used to analyze categorical variables in GraphPad (https://www.graphpad.com/quickcalcs/contingency1/, accessed on 10 June 2025). *p*-values less than 0.05 were deemed statistically significant.

## 3. Results

### 3.1. Characteristics of the Tested Group

This study was conducted over 4 years, during which 569 samples from children up to 6 years of age were tested for *S. pneumoniae*. The distribution of children by age group was as follows: under 12 months (73), 12–24 months (107), 36–59 years (220), and 5–6 years (169). PCR testing detected *S. pneumoniae* in nasopharyngeal swabs from 232 children, indicating a carrier rate of 40.8% in the study population. In comparison, the traditional microbiological method isolated *S. pneumoniae* from only 26 nasal secretions, representing only a 4.6% positivity rate (*p* = 0.0001). The carrier rates for both sexes were similar, with 128 males (42.2%) and 105 females (39.4%) testing positive for *S. pneumoniae*.

### 3.2. S. pneumoniae Serotypes

We conducted an analysis to differentiate between capsular and non-capsular strains of *S. pneumoniae*. Our findings revealed that capsular strains accounted for 214 (91.8%) of the samples, while only 18 (7.7%) were non-capsular strains. We then analyzed the capsular *S. pneumoniae* to identify their serogroups/serotypes. The kit we used for this analysis successfully differentiated *S. pneumoniae* in 168 samples (78.6%) from children under 6 years of age, with 46 samples (21.4%) remaining untyped. The following *S. pneumoniae* serotypes were present at the highest percentages: 19B/19C (*n* = 40; 17.2%), 6C (*n* = 25; 10.7%), 15B/15C (*n* = 23; 9.8%), 19A (*n* = 17; 7.3%), 23B (*n* = 16; 6.9%), 11A/D (*n* = 15; 6.4%) (Figure 1).

### 3.3. Vaccine Serotypes

Out of the 255 serotypes detected, 91 (15.9%) were covered by at least one of the available vaccines: PCV7, PCV10, PCV13, PCV15, or PCV20. These 91 serotypes belonged to 14 different serotype/serogroup categories. However, 164 serotypes, representing 25.4% of 22 different categories, were not included in these vaccines. Specifically, 6% of the documented pneumococcal carriage was attributed to serotypes 3, 19A, and 6A, which are included as vaccine strains in PCV13 but not covered by PCV10. The carriage rate of the strains included in the PCV10 vaccine was only 1% (refer to Table 1 for further details).

### 3.4. The Dynamics of the Spread of S. pneumoniae

In the first year of this study (2021), we found that the prevalence of *S. pneumoniae* carriage among children under 6 years of age was 34.3%. In the following year (2022), we observed a decrease in the frequency of this carriage (27%). The decrease was even more significant in 2023, with the carriage frequency reducing to 22%, a statistically significant change compared with the 2021 data (*p* < 0.05). In 2024, the presence of *S. pneumoniae* in the nasopharynx of children increased to 47% among those surveyed. This increase was statistically significant compared with the levels observed in 2022 and 2023 (*p* < 0.05). All 54 healthy children studied in the first 3 months of 2025 were found to be carriers of *S. pneumoniae*.

#### 3.4.1. The Dynamics of the Spread of *S. pneumoniae* Serotypes

A total of 24 different *S. pneumoniae* serotypes were identified in 2021. By 2024, this number had decreased to 18. The years 2022 and 2023 showed less diversity, with 17 and 14 serotypes identified, respectively. In the first half of 2025, 16 different serotypes of *S. pneumoniae* were identified in children under 6 years of age. The predominant serotype from 2021 to 2024 was 19B/19C, with a detection rate of 7–9%. The most common serotype in 2025 was 19A, with a detection rate of 19%, followed by 6C, 15B/15C, and 11A/D, with detection rates of 17.6%, 17.6%, and 11.8%, respectively, among children tested for *S. pneumoniae* carriage (Figure 2).

Analysis of the variation in the detected serotypes among the children tested was conducted in relation to their presence in previous years. In 2021, eight of the detected serotypes (14, 21, 31, 19F, 39F, 41A/41F, and 9A/9V) were unique, as they were not observed in children tested in previous years. In 2022, we detected serotypes 24A, 24B/24F, 33F/33A, 6D, and 9N/9L in this group of children, noting varying rates of detection in subsequent years. In 2023, newly identified serotypes included 4, 5, 38, 35C, and 35F. The newly identified serotype in 2024 was 22F/22A; this was also identified in the group of children tested in 2025.

#### 3.4.2. Age Distribution of Detected *S. pneumoniae* Serotypes

Participants in this study were divided into four age groups to examine the relationship between age and *S. pneumoniae* carriage (Table 2). Children under 12 months of age were less likely to be carriers of *S. pneumoniae* (20.4%) compared with the other three age groups (12–35 months, 38.08%; 36–59 months, 71.89%; 5–6 years, 76.38%). Non-capsulated *S. pneumoniae* was significantly less common in children under 12 months of age (5.6%) compared with encapsulated *S. pneumoniae* in children aged 36 months to 6 years (36–59 months, 33%; 5–6 years, 43%; *p* < 0.05). Furthermore, the incidence of capsulated *S. pneumoniae* was significantly lower in children under 12 months of age (14.8%) compared with the incidence of non-encapsulated *S. pneumoniae* in children aged 36 months to 6 years (36–59 months, 38.9%; 5–6 years, 33.3%; *p* < 0.05).

Serotype 6C was detected more frequently in children under 12 months of age than in the other three age groups, with a prevalence of 25% compared with 15% in the 12–35 months group (n = 0.0996), 15% in the 36–59 months group (n = 0.2318), and 8.5% in the 5–6 years group (n = 0.0398) (see Table 1). Serogroup 10A also showed a higher detection rate in the youngest children, at 14.3%, compared with none detected in the 12–35 months group (n = 0.0267) and the 36–59 months group (n = 0.0088), and 5% in the 5–6 years group (n = 0.2051) (see Table 1). There was a predominance of *S. pneumoniae* serogroups 19B/19C and 7C/40 in the 36–59 months and 5–6 years age groups compared with the younger children. A significant difference was observed in the frequency of 19B/19C carriage in children over 36 months of age compared with children in the 12–35 months group (10.7%), with a prevalence of 26% in the 36–59 months group and 32.2% in the 5–6 years group (*p* < 0.05).

### 3.5. Mono- and Co-Detection of S. pneumoniae Serotypes

In this study involving 569 children under the age of 6 years, we observed both mono- and co-carriage of multiple serotypes of *S. pneumoniae*. Specifically, one serotype was detected in 116 children, accounting for 20.4% of the total number of children. In contrast, more than one serotype was identified in 52 children, representing 9.1% of the group. The distribution of detected serotypes among the patients was as follows: two serotypes were found in 30 (17.9%) children, three serotypes in 15 (8.9%) children, four serotypes in 5 (2.9%) children, five serotypes in 1 (0.6%) child, and six serotypes in 1 (0.6%) child. Figure 3 shows that, among children screened for *S. pneumoniae* carriage, some serotypes were more frequently detected together with other *S. pneumoniae* serotypes than alone. Serotypes 19B/19C, 15B/15C, 10B, 10F/C, 23B, 7C/40, 23A, and 24A had co-detection rates of 55.8%, 28.9%, 21.2%, 21.2%, 17.3%, 15.4%, and 13.4%, respectively. In contrast, the mono-detection rates of these serotypes were significantly lower: 9.5%, 6.9%, 0.9%, 0.9%, 4.3%, 0.9%, 3.5%, and 0.9%. The difference was statistically significant (*p* < 0.05). Serotypes 14, 20, 31, 38, 22F/22A, 23F, 33F/33A, and 39F were detected individually, with no accompanying serotypes.

This study found that the highest rate of co-carriage of multiple serotypes occurred in children aged 36 to 60 months, with a rate of 39.4%. In contrast, only 17.9% of children under 35 months had multiple serotypes; this difference was statistically significant (*p* = 0.0030). Among infants under 12 months of age, a small proportion (17.9%) were carriers of two serotypes, while in older age groups, the co-carriage of three, four, five, or even six serotypes of *S. pneumoniae* was observed (see Table 3).

## 4. Discussion

Invasive pneumococcal disease continues to represent a significant global health challenge, even in developed countries, despite the introduction of the PCV vaccine. This study investigated the prevalence of *S. pneumoniae* over a 10-year period after the introduction of the PCV10 vaccine, which was added to the immunization schedule in Bulgaria. The results provide information on the effects of vaccination and other factors on nasopharyngeal colonization, changes in serotypes, and antimicrobial susceptibility of pneumococci.

*S. pneumoniae* is commonly found in the human nasopharynx, with prevalence ranging from 27% to 65% in children [24]. In our study, we observed a prevalence of 40.8%, which falls within this range. The high carriage rates may be due to the fact that many of the reported *S. pneumoniae* serotypes are not included in the PCV10 vaccine, leading to an increase in the total number of detected cases [25]. The predominant serotypes identified in our study include 19B/19C, 6C, 15B/15C, 19A, 23B, and 11A/D. A prospective study conducted in Marrakech, Morocco, between 2017 and 2018 followed healthy children attending vaccination centers. That study revealed an increased prevalence of serotypes 19B/19C and 15B/15C among these children [26]. The vaccination program implemented during this time is similar to that used in Bulgaria and involves the administration of PCV10. Additionally, a study conducted in infants in Belgium reported an increase in the prevalence of serotypes 6C and 19A during the transition period from PCV10 vaccination to PCV13 [27].

A Belgian study found that the increase in certain pneumococcal serotypes is linked to the vaccination strategy and the return to the use of the PCV10 vaccine [28]. This approach is similar to Bulgaria’s vaccination strategy, which also utilizes the PCV10 vaccine, despite it not covering specific serotypes. Notably, an increase in serotype 19A was observed after the switch to PCV10, particularly among children with invasive pneumococcal disease (IPD) [29]. This led to an increase in pediatric cases starting in 2017. By 2020, serotype 19A represented 19.3% of cases in Belgium, marking an increase of 3.8% compared with 2019 [27].

Recent studies have shown a worrying increase in the prevalence of serotype 6C in Bulgaria [30], mirroring findings from other countries [27,31]. In Belgium, for example, serotype 6C is implicated in 5.9% of IPD cases [29]. In our study, the prevalence of serotype 6C among children under 6 years of age was 10.7%. This increase was associated with the use of the pneumococcal conjugate vaccine PCV10 instead of PCV13 in Bulgaria. Although PCV13 does not directly target serotype 6C, its prevalence appears to be associated with serotype 6A, which is included in PCV13, possibly due to a presumed cross-protection between these two serotypes [32]. In addition, cases of tetracycline and erythromycin resistance have been documented alongside the increasing prevalence of serotype 6C [33,34].

In this follow-up study, lower levels of *S. pneumoniae* carriage were recorded in 2022 and 2023 compared with 2021 and 2024. This decrease may be due to non-pharmaceutical measures introduced during the COVID-19 pandemic, such as the closure of childcare facilities and the reduction in close contact between people [35]. A study in England reported that the overall incidence of invasive lung disease was lower in 2022–2023 than in 2019–2020 [36]. Therefore, after the lifting of these measures in 2022 and the official end of the pandemic in 2024, a significant increase in recorded carriage cases was observed. The diversity of identified serotypes reflects this pattern. While serotype diversity was noted in 2021, a slight decrease in diversity was observed in 2022 and 2023; however, we again observed an increase in the diversity of *S. pneumoniae* serotypes in 2024 and early 2025. There was no clear temporal relationship between serotypes, as they disappeared and then reappeared. For example, from 2013 to 2015, a predominance of serotypes 22F/22A was reported in Madrid [37]. However, these serotypes were not observed in Bulgaria until 2024. Thus, the distribution of the different serotypes shows different geographical characteristics. The prevalence of invasive serotypes of *S. pneumoniae* also varies geographically [38]. Colonization rates are higher among children during the fall and winter months, which coincides with an increase in colds and viral upper respiratory tract infections, often accompanied by nasal discharge [39].

*S. pneumoniae* shows age-related preferences in terms of carriage rates, influenced by geographic and seasonal factors. Studies have shown that children who are carriers play a key role in the transmission of pneumococci to the elderly. Approximately 10% of adults are asymptomatic carriers [40]. Noteworthily, children under 2 years of age have a lower rate of carriage compared to those aged 3 to 6 years. This difference may be due to the higher frequency of attendance at daycare and preschool compared with children under 3 years of age. Factors that increase the likelihood of asymptomatic carriage in children include attendance at daycare or school and the number of children living in the household [41]. While pneumococcal infections can affect people of all ages, children under two years of age and adults over 65 years of age are at significantly higher risk of developing complications [42]. A relationship has been established between the age group of the children and the prevalence of different serotypes. A study conducted in 2017 and 2018 found that serotypes 6C and 11A/D were more prevalent in children under 12 months of age [26], which aligns with our findings. However, we cannot confirm our conclusion that higher carriage of serotypes 19B/19C was observed in children over 36 months of age compared with younger children, as this is not supported by other studies.

Monitoring nasopharyngeal carriage in children hospitalized with clinical pneumonia revealed that half of the patients carried more than one serotype of *S. pneumoniae* simultaneously [43]. In contrast, our study found that only 9.1% of the examined healthy children carried multiple serotypes. This difference may be attributed to the fact that our study focused on healthy children, whereas the cited study concentrated on hospitalized patients. Thus, we can conclude that the carriage of multiple serotypes is more common among hospitalized children than among healthy children. Research has demonstrated that introducing vaccines with a higher number of vaccine serotypes leads to a reduction in the carriage of multiple serotypes [44]. We frequently detected serotypes 19B/19C together with other serotypes. Additionally, we identified other co-detected serotypes, including 10B, 10F/C, 23B, 7C/40, 23A, and 24A, that are not currently included in any of the available vaccines. This emphasizes the importance of creating new generations of vaccines that incorporate these co-detected serotypes. It is expected that addressing these serotypes will help reduce the incidence of pneumonia caused by non-vaccine pneumococcal serotypes.

The epidemiological results of this study show an increase in the simultaneous detection of more than two serotypes in children aged 5 to 6 years. This finding highlights the need to develop a vaccine that covers a greater percentage of non-vaccine serotypes. However, this study has some limitations, such as its narrow scope and including children from only 15 of the 28 regions in Bulgaria. Additionally, it is crucial to investigate the presence of *S. pneumoniae* in hospitalized patients of all ages to validate our findings regarding the effectiveness of the country’s vaccination strategy. We plan to conduct a future study focusing on hospitalized patients with respiratory symptoms to assess the carriage of *S. pneumoniae*. Furthermore, we did not analyze the antibiotic resistance of individual serotypes because we could not isolate them using conventional microbiological methods. To address this gap, we intend to conduct sequencing analysis in the future, which will help determine if any antibiotic resistance genes are present in the strains we encounter.

## 5. Conclusions

This review reveals a relatively high prevalence of *S. pneumoniae* carriage among healthy children up to 6 years of age who were vaccinated with PCV10. The study identified the predominant serotypes as 6C, 15B/15C, and 19A, while serotype 19B/19C was found to be the most common cause of both single and multiple nasopharyngeal colonization in children. Furthermore, children over 3 years of age were more likely to be colonized with more than two pneumococci. These findings highlight the need to review the childhood vaccination strategy. Such a change is essential to reduce the emergence and spread of antibiotic-resistant serotypes both in our country and in neighboring regions

## Figures and Tables

**Figure 1 vaccines-13-00634-f001:**
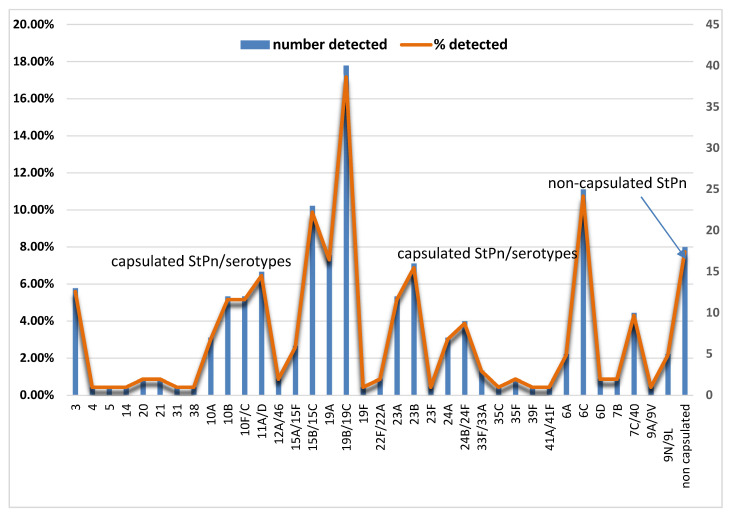
The distribution of detected encapsulated and non-encapsulated *Streptococcus pneumoniae*. The distribution of the type, number, and percentage of non-encapsulated *S. pneumoniae* is presented, along with serotypes of encapsulated *S. pneumoniae* (StPn*).

**Figure 2 vaccines-13-00634-f002:**
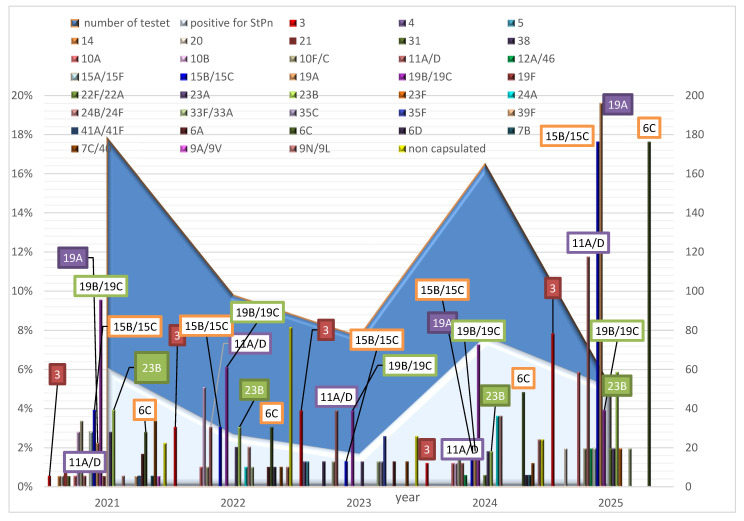
The dynamics of the spread of non-capsular and capsular *S. pneumoniae*, along with their serotypes, for each year from 2021 to 2025 among healthy children under 6 years of age.

**Figure 3 vaccines-13-00634-f003:**
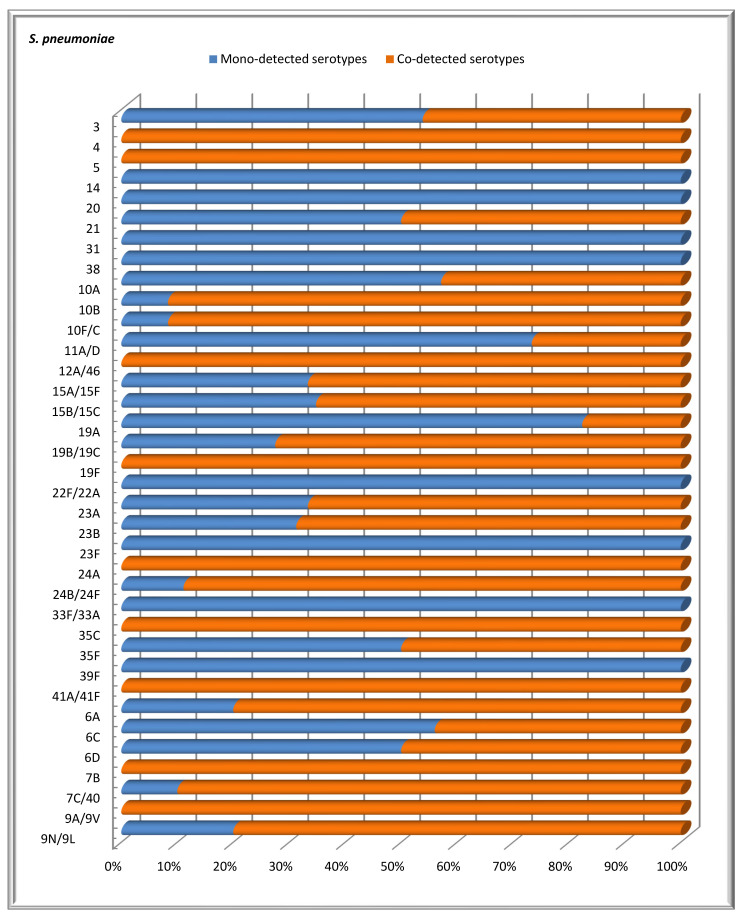
The distribution of confirmed *S. pneumoniae* serotypes based on the proportions of mono- and co-detections among healthy children under 6 years of age.

**Table 1 vaccines-13-00634-t001:** The distribution of vaccine serotypes in different vaccine generations compared with *Streptococcus pneumoniae* serotypes detected in children under 6 years of age.

StPn Serotypes	Number of Detections Out of a Total of 569 Patients Tested	Presence in Vaccine (Vaccine Generation)
3	13 (2.3)	PCV13, PCV15, PCV20
4	1 (0.2)	PCV7; PCV10, PCV13, PCV15, PCV20
5	1 (0.2)	PCV10, PCV13, PCV15, PCV20
14	1 (0.2)	PCV7; PCV10, PCV13, PCV15, PCV20
20	2 (0.4)	not present
21	2 (0.4)	not present
31	1 (0.2)	not present
38	1 (0.2)	not present
10A	7 (1.2)	PCV20
10B	12 (2.1)	not present
10F/C	12 (2.1)	not present
11A/D	15 (2.6)	PCV20 (11A)
12A/46	2 (0.4)	not present
15A/15F	6 (1.1)	not present
15B/15C	23 (4)	PCV20 (15B)
19A	17 (2.9)	PCV13, PCV15, PCV20
19B/19C	40 (7)	not present
19F	1 (0.2)	PCV7; PCV10, PCV13, PCV15, PCV20
22F/22A	2 (0.35)	PCV15, PCV20 (22F)
23A	12 (2.1)	not present
23B	16 (2.8)	not present
23F	1 (0.2)	PCV7; PCV10, PCV13, PCV15, PCV20
24A	7 (1.2)	not present
24B/24F	9 (1.5)	not present
33F/33A	3 (0.5)	PCV15, PCV20 (33F)
35C	1 (0.2)	not present
35F	2 (0.35)	not present
39F	1 (0.2)	not present
41A/41F	1 (0.2)	not present
6A	5 (0.9)	PCV13, PCV15, PCV20
6C	25 (4.4)	not present
6D	2 (0.35)	not present
7B	2 (0.35)	not present
7C/40	10 (1.8)	not present
9A/9V	1 (0.2)	PCV7; PCV10, PCV13, PCV15, PCV20 (9V)
9N/9L	5 (0.9)	not present

**Table 2 vaccines-13-00634-t002:** Distribution of detected *S. pneumoniae* serotypes in different age groups: under 12 months, 12–35 months, 36–59 months, and 5–6 years.

Capsulated StPn/Serotypes	<12 mo, *n*(%) (Total *n* = 28)	12–35 mo, *n*(%) (Total *n* = 39)	36–59 mo *n*(%) (Total *n* = 60)	5–6 Years (Total *n* = 59)
3	3 (10.7)	1 (2.56)	3 (5)	6 (10.2)
4	0 (0)	1 (2.56)	0 (0)	0 (0)
5	0 (0)	1 (2.56)	0 (0)	0 (0)
14	0 (0)	0 (0)	0 (0)	1 (1.7)
20	0 (0)	0 (0)	1 (1.7)	1 (1.7)
21	0 (0)	0 (0)	2 (3.3)	0 (0)
31	0 (0)	0 (0)	1 (1.7)	0 (0)
38	0 (0)	1 (2.56)	0 (0)	0 (0)
10A	4 (14.3)	0 (0)	0 (0)	3 (5)
10B	0 (0)	1 (2.56)	5 (8.3)	6 (10.2)
10F/C	1 (3.6)	0 (0)	2 (3.3)	9 (15.3)
11A/D	3 (10.7)	5 (12.8)	5 (8.3)	2 (3.4)
12A/46	1 (3.6)	1 (2.56)	0 (0)	0 (0)
15A/15F	0 (0)	1 (2.56)	2 (3.3)	3 (5)
15B/15C	3 (10.7)	6 (15.4)	9 (15)	5 (8.5)
19A	1 (3.6)	6 (15.4)	4 (6.7)	6 (10.2)
19B/19C	3 (10.7)	2 (5.1)	16 (26.7)	19 (32.2)
19F	0 (0)	0 (0)	0 (0)	1 (1.7)
22F/22A	1 (3.6)	1 (2.56)	0 (0)	0 (0)
23A	0 (0)	1 (2.56)	6 (10)	5 (8.5)
23B	0 (0)	5 (12.8)	7 (11.6)	4 (6.8)
23F	1 (3.6)	0 (0)	0 (0)	0 (0)
24A	1 (3.6)	1 (2.56)	3 (5)	2 (3.4)
24B/24F	1 (3.6)	1 (2.56)	3 (5)	4 (6.8)
33F/33A	1 (3.6)	1 (2.56)	1 (1.7)	0 (0)
35C	1 (3.6)	0 (0)	0 (0)	0 (0)
35F	0 (0)	0 (0)	1 (1.7)	1 (1.7)
39F	0 (0)	0 (0)	0 (0)	1 (1.7)
41A/41F	0 (0)	0 (0)	1 (1.7)	0 (0)
6A	1 (3.6)	0 (0)	1 (1.7)	3 (5)
6C	7 (25)	4 (10.3)	9 (15)	5 (8.5)
6D	0 (0)	0 (0)	2 (3.3)	0 (0)
7B	0 (0)	0 (0)	0 (0)	2 (3.4)
7C/40	1 (3.6)	0 (0)	4 (6.7)	5 (8.5)
9A/9V	0 (0)	0 (0)	0 (0)	1 (1.7)
9N/9L	0 (0)	2 (5.1)	0 (0)	3 (5)

**Table 3 vaccines-13-00634-t003:** The age distribution of co-detected serotypes of *S. pneumoniae* among healthy children in different age groups: under 12 months, 12–35 months, 36–59 months, and 5–6 years.

Co-Detected Serotypes	Serotypes	<12 mo, *n*(%) (Total *n* = 28)	12–35 mo, *n*(%) (Total *n* = 39)	36–59 mo *n*(%) (Total *n* = 60)	5–6 Years *n*(%) (Total *n* = 59)
Co-detected with two serotypes	10A;12A/46	1 (3.6)	0 (0)	0 (0)	0 (0)
Co-detected with two serotypes	10B;19B/19C	0 (0)	0 (0)	2 (3.3)	0 (0)
Co-detected with two serotypes	10F/C;15B/15C	0 (0)	0 (0)	0 (0)	1 (1.69)
Co-detected with two serotypes	10F/C;19B/19C	0 (0)	0 (0)	0 (0)	3 (5)
Co-detected with two serotypes	15A/15F;19B/19C	0 (0)	0 (0)	0 (0)	1 (1.69)
Co-detected with two serotypes	15B/15C;19B/19C	0 (0)	0 (0)	2 (3.3)	0 (0)
Co-detected with two serotypes	19A;21	0 (0)	0 (0)	1 (1.7)	0 (0)
Co-detected with two serotypes	19A;6C	1 (3.6)	0 (0)	0 (0)	0 (0)
Co-detected with two serotypes	19B/19C;15A/15F	0 (0)	0 (0)	0 (0)	1 (1.69)
Co-detected with two serotypes	19B/19C;24B/24F	0 (0)	0 (0)	1 (1.7)	0 (0)
Co-detected with two serotypes	19B/19C;7C/40	0 (0)	0 (0)	0 (0)	1 (1.69)
Co-detected with two serotypes	23A;12A/46	0 (0)	1 (2.6)	0 (0)	0 (0)
Co-detected with two serotypes	23A;19B/19C	0 (0)	0 (0)	0 (0)	1 (1.69)
Co-detected with two serotypes	23A;23B	0 (0)	0 (0)	1 (1.7)	0 (0)
Co-detected with two serotypes	23B;15B/15C	0 (0)	1 (2.6)	2 (3.3)	0 (0)
Co-detected with two serotypes	3;10B	0 (0)	0 (0)	0 (0)	2 (3.4)
Co-detected with two serotypes	3;11A/D	0 (0)	0 (0)	0 (0)	1 (1.69)
Co-detected with two serotypes	3;15B/15C	0 (0)	0 (0)	1 (1.7)	0 (0)
Co-detected with two serotypes	3;19B/19C	1 (3.6)	0 (0)	0 (0)	0 (0)
Co-detected with two serotypes	6A;19B/19C	1 (3.6)	0 (0)	0 (0)	0 (0)
Co-detected with two serotypes	6C;10B	0 (0)	0 (0)	0 (0)	1 (1.69)
Co-detected with two serotypes	6C;10F/C	0 (0)	0 (0)	0 (0)	1 (1.69)
Co-detected with two serotypes	6C;15A/15F	0 (0)	1 (2.6)	0 (0)	0 (0)
Co-detected with two serotypes	6C;15B/15C	0 (0)	0 (0)	1 (1.7)	1 (1.69)
Co-detected with two serotypes	6C;19A	0 (0)	0 (0)	0 (0)	1 (1.69)
Co-detected with two serotypes	6C;19B/19C	0 (0)	0 (0)	2 (3.3)	0 (0)
Co-detected with two serotypes	6D;7C/40	0 (0)	0 (0)	1 (1.7)	0 (0)
Co-detected with two serotypes	7C/40;19B/19C	0 (0)	0 (0)	0 (0)	1 (1.69)
Co-detected with two serotypes	7C/40;35C	1 (3.6)	0 (0)	0 (0)	0 (0)
Co-detected with two serotypes	7B;7B/40	0 (0)	0 (0)	0 (0)	1 (1.69)
Co-detected with three serotypes	11A/D;24A;24B/24F	0 (0)	0 (0)	1 (1.7)	0 (0)
Co-detected with three serotypes	19B/19C;24A;24B/24F	0 (0)	0 (0)	1 (1.7)	1 (1.69)
Co-detected with three serotypes	23A;10F;C19B/19C	0 (0)	0 (0)	1 (1.7)	0 (0)
Co-detected with three serotypes	9N/9L;19B/19C	0 (0)	0 (0)	0 (0)	1 (1.69)
Co-detected with three serotypes	7C/40;24A;24B/24F	0 (0)	0 (0)	1 (1.7)	0 (0)
Co-detected with three serotypes	24A;24B/24F;9N/9L	0 (0)	0 (0)	0 (0)	1 (1.69)
Co-detected with three serotypes	10F/C;7C/40;19B/19C	0 (0)	0 (0)	1 (1.7)	0 (0)
Co-detected with three serotypes	10F/C;7C/40;7B	0 (0)	0 (0)	0 (0)	1 (1.69)
Co-detected with three serotypes	15B/15C;10A;10B	0 (0)	0 (0)	0 (0)	1 (1.69)
Co-detected with three serotypes	23B;10B;15B/15C	0 (0)	0 (0)	1 (1.7)	2 (3.4)
Co-detected with three serotypes	23B;19B/19C;35F	0 (0)	0 (0)	0 (0)	1 (1.69)
Co-detected with three serotypes	3;23A;11A/11D	0 (0)	0 (0)	1 (1.7)	0 (0)
Co-detected with three serotypes	6A;10F/C;19B/19C	0 (0)	0 (0)	0 (0)	1 (1.69)
Co-detected with three serotypes	6A;15A/15F;19B/19C	0 (0)	0 (0)	0 (0)	1 (1.69)
Co-detected with three serotypes	9N/9L;10F/C;11A/D	0 (0)	0 (0)	0 (0)	1 (1.69)
Co-detected with four serotypes	19B/19C;24A;24B/24F;15B/15C	0 (0)	0 (0)	1 (1.7)	0 (0)
Co-detected with four serotypes	23A;23B;10B;15B/15C	0 (0)	0 (0)	1 (1.7)	0 (0)
Co-detected with four serotypes	19F;23A;9A/9V;19B/19C	0 (0)	0 (0)	0 (0)	1 (1.69)
Co-detected with four serotypes	23B;10A;10F/C;19B/19C	0 (0)	0 (0)	0 (0)	1 (1.69)
Co-detected with four serotypes	6A;7C/40;19B/19C;41A/41F	0 (0)	0 (0)	1 (1.7)	0 (0)
Co-detected with five serotypes	6C;23A;23B;10B;15B/15C	0 (0)	1 (2.6)	0 (0)	0 (0)
Co-detected with six serotypes	4;5;6C;9N/9L;24A;24B/24F	0 (0)	1 (2.6)	0 (0)	0 (0)

## Data Availability

https://grippe.gateway.bg/page.php?category=77 accessed on 10 June 2025.

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
