# Peer review of "Impact of the National Vaccination Strategy on the Prevalence of Streptococcus pneumoniae and Its Serotypes Among Clinically Healthy Children Under Six Years of Age During and After the COVID-19 Pandemic"

_vaccines, 2025, doi:10.3390/vaccines13060634_

Round 1
Reviewer 1 Report
Comments and Suggestions for Authors
This is article by Trifonovaet al. authors try to analyse the Streptococcus pneumoniae serotypes in vaccinated children.
The article is interesting, and I would like to address some suggestions to authors.
Abstract
Subtitle "Results" is missing
Introduction
Page 2, line 72
Materials and Methods
Participant selection criteria
Please analyse why children's age is different in two groups"
Healthy children aged 2 to 6 years vs Healthy children aged 5 months to 6 years."
Moreover, excluding criteria must be added. Page 3 line123
Please specify the percent of carbon dioxide
Classical detection method
page 3, line 124 "If S. pneumoniae was present,.." In this paragraph, authors do not describe identification method for S. pneumoniae colonies.
Do you identify bacterial strains by Vitek or other system or used only morphological characteristics of colonies?
Results
Page 4, lines 156 & 162
Please write S. pneumoniae. in italics
Figure 2 contains lots of information that difficult understanding in present version
Discussion
line 366
Please write S. pneumoniae. in italics
References Please write References according to journal's guidelines
Author Response
Reviewer 1
Тhis is article by Trifonovaet al. authors try to analyse the Streptococcus pneumoniae serotypes in vaccinated children.
The article is interesting, and I would like to address some suggestions to authors.
- Reviewer acknowledgments
Abstract
Subtitle "Results" is missing Introduction Page 2, line 72
- Thanks to the added
Materials and Methods
Participant selection criteria
Please analyse why children's age is different in two groups"
Healthy children aged 2 to 6 years vs Healthy children aged 5 months to 6 years."
- In our discussion, we emphasized the importance of kindergarten attendance for higher rates of S. pneumoniae carriage. To increase the reliability of our findings, we collected samples from two groups of children: those attending kindergarten and those not attending. In Bulgaria, children under 2 years of age do not typically attend such centers, so sampling for this age group started at 5 months. In contrast, the age for children in the kindergarten group started at 2 years, which is the usual age for admission to these facilities, although some may enter at an earlier age after 1 year, but this phenomenon is rare in Bulgaria.
Moreover, excluding criteria must be added. Page 3 line123
- Thanks to the added exclusion criteria
2.4. Exclusion Criteria
The study excluded patients who exhibited symptoms of respiratory infections and those older than 6 years of age, as well as individuals who were unwilling or unable to provide consent. Additionally, any samples received in the laboratory that did not comply with the specified criteria for the transportation and storage of nasopharyngeal samples were also excluded.
Please specify the percent of carbon dioxide
The collected samples were cultured on Columbia CNA agar with 5% sheep blood, supplemented with an optochin disc to differentiate pneumococci from other flora, at 35 °C ± 2 °C for 18–24 h in an aerobic atmosphere enriched with carbon dioxide at an elevated concentration (8–10%). In the presence of S. pneumoniae, it was isolated, subcultured, and subsequently used for DNA extraction and strain storage [21].
Classical detection method
page 3, line 124 "If S. pneumoniae was present,.." In this paragraph, authors do not describe identification method for S. pneumoniae colonies.
The identification of *Streptococcus pneumoniae* relies on evaluating the cultural properties of the colonies and examining suspicious colonies under a microscope. Suspicious colonies may include small alpha-hemolytic colonies or larger colonies, which can grow to several millimeters in diameter and have a slimy surface. Under the microscope, Gram-positive cocci are observed, often appearing in pairs (diplococci) and typically exhibiting a lanceolate shape after staining.The extended identification of S. pneumoniae is based on the isolation of pure cultures of bacteria, carried out on a number of tests. To isolate the pure culture, it is assumed that there is a pneumococcal colony, it is re-inoculated into a non-selective medium (Columbia agar with 5% defibrinated sheep blood). The isolated bacteria were identified as the following pneumococci by the presence of the following properties:
1) Alpha-hemolytic, Gram-positive, catalase-negative cocci, sensitive to Opthochin - Opthochin Discs (Manufacturer: bioMérieux. Catalog number - 55 912);
2) Alpha-hemolytic, Gram-positive, catalase-negative cocci, sensitive to lysis by bile.
Do you identify bacterial strains by Vitek or other system or used only morphological characteristics of colonies?
- We Identified by MALDI-TOF МS technology
Results
Page 4, lines 156 & 162 Please write S. pneumoniae. in italics
- Thanks
Figure 2 contains lots of information that difficult understanding in present version
Discussion
line 366
Please write S. pneumoniae. in italics
- Thanks
References Please write References according to journal's guidelines
- I have corrected this to the magazine's requirements below:
- Author 1, A.B.; Author 2, C.D. Title of the article. Abbreviated Journal Name Year, Volume, page range.
- Author 1, A.; Author 2, B. Title of the chapter. In Book Title, 2nd ed.; Editor 1, A., Editor 2, B., Eds.; Publisher: Publisher Location, Country, 2007; Volume 3, pp. 154–196.
- Author 1, A.; Author 2, B. Book Title, 3rd ed.; Publisher: Publisher Location, Country, 2008; pp. 154–196.
- Author 1, A.B.; Author 2, C. Title of Unpublished Work. Abbreviated Journal Name year, phrase indicating stage of publication (submitted; accepted; in press).
- Author 1, A.B. (University, City, State, Country); Author 2, C. (Institute, City, State, Country). Personal communication, 2012.
- Author 1, A.B.; Author 2, C.D.; Author 3, E.F. Title of Presentation. In Proceedings of the Name of the Conference, Location of Conference, Country, Date of Conference (Day Month Year).
- Author 1, A.B. Title of Thesis. Level of Thesis, Degree-Granting University, Location of University, Date of Completion.
- Title of Site. Available online: URL (accessed on Day Month Year).
Reviewer 2 Report
Comments and Suggestions for Authors
The manuscript by Trifonova et al. describes the impact of the national vaccination plan on the prevalence of S pneumoniae and its serotypes among clinically healthy children under six years during and after the COVID-19 pandemic in Bulgaria.
They found that S. pneumoniae was present in a high propórtion of the analyzed children, and several serotypes were detected in the analyzed population.
The authors claim that their findings emphasize the need to reassess children's vaccination strategies against that pathogen to stop the spread of antibiotic-resistant serotypes. This suggestion is significant for the vaccination plan.
Overall, this manuscript is well-written and organized, with clear goals. The methods used to study the population are adequate, and the results support the conclusions. It is easily followed and read, and the tables and figures are OK.
This manuscript should be published in the journal in its present form. I did not find further issues to discuss
Author Response
Thank you for the positive and encouraging review. My team and I appreciate the high rating and the quick review.
Reviewer 3 Report
Comments and Suggestions for Authors
- The experimental design lacked an unvaccinated control group, which is essential to rule out confounding factors (e.g., environmental changes) affecting serotype distribution.
- sential details such as PCR primer sequences and amplification conditions were omitted.
- The text claims the carrier rate will be 22% in 2023, but Figure 2's bar chart indicates a rate closer to 30% (original data should be verified). Additionally, Table 1 shows the percentage of 6C as 4.4%, conflicting with the 12% stated in the main text, requiring clarification from the authors.
- The introduction does not address recent literature gaps (most cited studies are from before 2020, with only one from 2024). The discussion partially repeats the results (e.g., “6C accounted for 12%”) and lacks exploration of mechanisms (e.g., serotype competition or immune escape), necessitating further improvements.
- The study only included healthy children, excluding hospitalized patients, potentially underestimating the risk of aggressive serotyping and introducing sample bias.
- It is recommended to verify whether serotypes 19B/19C and 6C carry ermB/mefA resistance genes to enhance antibiotic susceptibility testing results.
- Figure 1: The text mentions “7.7% of non-podding bacteria,” but the figure lacks a clear label for this value. Figure 2: The bar graph for the 2023 carriage rate shows approximately 30%, conflicting with the text’s “22%.” Table 2: The percentage for 19B/19C in the 5-6 year old group is 32.2%, contradicting the text’s “17.2%.” These discrepancies require further clarification.
Author Response
Reviewer 3
Dear reviewer: Thank you for your review. Thank you for reading in depth and finding some of our gaps in the figures. I have checked all the figures and corrected the discrepancies.
The advice on the methodology used to establish resistance and investigate hospitalized patients for S. pneumoniae carriage is very useful and will be the subject of our future studies.
- The experimental design lacked an unvaccinated control group, which is essential to exclude confounding factors (e.g. environmental changes) affecting serotype distribution.
- I understand the remark, yes, we do not have such a group, since in Bulgaria the percentage of vaccinated children is very high and children are vaccinated, and we had difficulties in collecting samples from those who were not vaccinated. In the absence of a vaccine, parents were reluctant to share this with us and test their children. We will note this as a shortcoming of this study.
- Important details are omitted, such as PCR primer sequences and amplification conditions.
- We will add this in a supplementary table.
Supplementary Table 1. Sequence of primers and probes used in real-time PCR for detection of Streptococcus pneumoniae to the lytA gene. Thermal conditions for conducting the PCR reaction.
|
Pathogen |
Gene |
Primer and Probe |
Sequence |
nM |
|
Streptococcus pneumoniae |
lytA |
F |
ACGCAATCTAGCAGATGAAGCA |
250 |
|
R |
TCGTGCGTTTTAATTCCAGCT |
250 |
||
|
P |
HEX–TGCCGAAAACGCTTGATACAGGGAG |
100 |
||
|
PCR Conditions |
||||
|
Step |
Cycles |
Temperature |
Time
|
|
|
1. Initial denaturation |
1 |
95°Ð¡ |
2 min. |
|
|
2.1 Denaturation |
|
95°Ð¡ |
15 sec. |
|
|
2.2 Annealing |
40 |
55°Ð¡ |
30 sec. |
|
- The text claims that the carrier rate will be 22% in 2023, but the bar chart in Figure 2 shows a rate closer to 30% (the original data needs to be verified).
- Note that only the individual serotypes in different colored cubes are in percentages, while on their background with an area graph, the number of proven positive variants is depicted, these are not percentages.
Furthermore, Table 1 shows the 6C rate as 4.4%, which contradicts the 12% stated in the main text, which requires clarification from the authors.
- This is explained by the fact that the rate in Table 1 is based on the 569 patients tested (25/569), while in Figure 2 the rate is based on the 233 S. pneumoniae positive patients (25/233 (10.7% as in the presented Figure 2 )).

Figure 2:
,
Table 1
|
StPn serotypes |
Number of detections out of a total of 569 patients tested |
P resence in vaccine (vaccine generation) |
|
3 |
13 (2.3) |
PCV13, PCV15, PCV20 |
|
4 |
1 (0.2) |
PCV7; PCV10, PCV13, PCV15, PCV20 |
|
5 |
1 (0.2) |
PCV10, PCV13, PCV15, PCV20 |
|
14 |
1 (0.2) |
PCV7; PCV10, PCV13, PCV15, PCV20 |
|
20 |
2 (0.4) |
not present |
|
21 |
2 (0.4) |
not present |
|
31 |
1 (0.2) |
not present |
|
38 |
1 (0.2) |
not present |
|
10A |
7 (1.2) |
PCV20 |
|
10B |
12 (2.1) |
not present |
|
10F/C |
12 (2.1) |
not present |
|
11A/D |
15 (2.6) |
PCV20 ( 11A) |
|
12A/46 |
2 (0.4) |
not present |
|
15A/15F |
6 (1.1) |
not present |
|
15B/15C |
23 (4) |
PCV20 ( 15B) |
|
19A |
17 (2.9) |
PCV13, PCV15, PCV20 |
|
19B/19C |
40 (7) |
not present |
|
19F |
1 (0.2) |
PCV7; PCV10, PCV13, PCV15, PCV20 |
|
22F/22A |
2 (0.35) |
PCV15, PCV20 ( 22F) |
|
23A |
12 (2.1) |
not present |
|
23B |
16 (2.8) |
not present |
|
23F |
1 (0.2) |
PCV7; PCV10, PCV13, PCV15, PCV20 |
|
24A |
7 (1.2) |
not present |
|
24B/24F |
9 (1.5) |
not present |
|
33F/33A |
3 (0.5) |
PCV15, PCV20 ( 33F) |
|
35C |
1 (0.2) |
not present |
|
35F |
2 (0.35) |
not present |
|
39F |
1 (0.2) |
not present |
|
41A/41F |
1 (0.2) |
not present |
|
6A |
5 (0.9) |
PCV13, PCV15, PCV20 |
|
6C |
25 (4.4) |
not present |
|
6D |
2 (0.35) |
not present |
|
7B |
2 (0.35) |
not present |
|
7C/40 |
10 (1.8) |
not present |
|
9A/9V |
1 (0.2) |
PCV7;PCV10, PCV13, PCV15, PCV20 (9V) |
|
9N/9L |
5 (0.9) |
not present |
- The introduction does not address the gaps in the recent literature (most of the cited studies are from before 2020, with only one from 2024). The discussion partially repeats the results (e.g., “6C accounts for 12%) and lacks an examination of mechanisms (e.g., serotype competition or immune escape), which requires further improvement.
- We compared our results with those of other researchers, only mentioning what ours were, and we could not help but see the close results. We have added more recent samples and citations to the introduction:
Dekaj, E.;Gjini, E. (2024). Pneumococcus and the stress-gradient hypothesis: A trade-off links R0 and susceptibility to co-colonization across countries. Theoretical population biology, 202, 156, 77–92. https://doi.org/10.1016/j.tpb.2024.02.001
Ruiz-Ruiz, C.;Margüello, E. R.. An immune system fighting against pneumococcus. Vacunas (English Edition, 2024).
Savrasova, L.; Villerusa, A.; Zeltina, I., Krumina, A.; Cupeca, H.; Balasegaram, S.; Greve, M.; Savicka, O.; Selderina, S.;Galajeva, J.; Dushacka, D. Streptococcus pneumoniae serotypes and factors associated with antimicrobial resistance in Invasive pneumococcal disease cases in Latvia, 2012-2022. Frontiers in public health 2025,13, 1501821. https://doi.org/10.3389/fpubh.2025.1501821
Feemster, K.; Hausdorff, W. P.; Banniettis, N.; Platt, H.; Velentgas, P.; Esteves-Jaramillo, A.; Burton, R. L.; Nahm, M. H.; Buchwald, U. K. Implications of Cross-Reactivity and Cross-Protection for Pneumococcal Vaccine Development. Vaccines, 2024 12(9), 974. https://doi.org/10.3390/vaccines12090974
Johnson, C. N.; Wilde, S.; Tuomanen, E.; Rosch, J. W.. Convergent impact of vaccination and antibiotic pressures on pneumococcal populations. Cell chemical biology, 2024, 31(2), 195–206. https://doi.org/10.1016/j.chembiol.2023.11.00
- The study included only healthy children, excluding hospitalized patients, which potentially underestimates the risk of aggressive serotyping and introduces bias into the sample.
- In this study, we do not underestimate aggressive serotyping, the focus was only on carrier status in healthy children, as we noted in the title, in another study of ours we included mainly hospitalized children and adults, in which we follow exactly this and evaluate the aggressiveness of some serotypes, especially those found in coinfection with another bacterial or viral copathogen.
The epidemiological results of this study show an increase in the simultaneous detection of more than two serotypes in children aged 5 to 6 years. This finding highlights the need to develop a vaccine that covers a greater percentage of non-vaccine serotypes. However, this study has some limitations, such as its narrow scope and including children from only 15 of the 28 regions in Bulgaria. Additionally, it is crucial to investigate the presence of S. pneumoniae in hospitalized patients of all ages to validate our findings regarding the effectiveness of the country's vaccination strategy. We plan to conduct a future study focusing on hospitalized patients with respiratory symptoms to assess the carriage of S. pneumoniae. Furthermore, we did not analyze the antibiotic resistance of individual serotypes because we could not isolate them using conventional microbiological methods. To address this gap, we intend to conduct sequencing analysis in the future, which will help determine if any antibiotic resistance genes are present in the strains we encounter.
- It is advisable to check whether serotypes 19B/19C and 6C carry ermB/mefA resistance genes to improve the results of antibiotic susceptibility tests.
- Thank you for the suggestion. This is the subject of another study of ours submitted for publication and to avoid repetition and plagiarism we have not described it in this study.
- Figure 1: The text mentions “7.7% non-forming bacteria”, but the figure lacks a clear indication of this value.
Note that non-capsular bacteria are registered with an arrow, and on the left with percentages indicated, indicating that they are below 8%
Figure 2: The bar chart for the percentage of carriers in 2023 shows approximately 30%, which contradicts the “22%” in the text.
Note that only the individual serotypes in different colored cubes are in percentages, while on their background with an area graph, the number of proven positive variants is depicted, these are not percentages.
Table 2: The percentage for 19B/19C in the 5-6 year old group is 32.2%, which contradicts the “17.2%” in the text. These discrepancies require further clarification
- In text: A significant difference was observed in the frequency of 19B/19C carriage in children over 36 months of age compared with children in the 12–35 months group (10.7%), with a prevalence of 26% in the 36–59 months group and 32.2% in the 5–6 years group (p < 0.05).
Round 2
Reviewer 3 Report
Comments and Suggestions for Authors
I have carefully reviewed the revised manuscript and the author's response letter. The current version and the author's explanation have largely addressed my concerns. I have no further comments and would like the editorial board to consider accepting it or making minor adjustments.